# SARS-CoV-2 and Companion Animals: Sources of Information and Communication Campaign during the COVID-19 Pandemic in Italy

**DOI:** 10.3390/vetsci10070426

**Published:** 2023-06-30

**Authors:** Andrea Laconi, Barbara Saracino, Eliana Fattorini, Giuseppe Pellegrini, Massimiano Bucchi, Lucia Bailoni, Alessandra Piccirillo

**Affiliations:** 1Department of Comparative Biomedicine and Food Science, University of Padua, Legnaro, 35020 Padua, Italy; 2Department of Political and Social Sciences, University of Bologna, 40125 Bologna, Italy; barbara.saracino@unibo.it; 3Department of Sociology and Social Research, University of Trento, 38122 Trento, Italy; eliana.fattorini@unitn.it (E.F.); pellegrini@observanet.it (G.P.);

**Keywords:** SARS-CoV-2, COVID-19, companion animals, sources of information, communication, veterinarian

## Abstract

**Simple Summary:**

The present survey-type investigation aimed at assessing the sources and the level of information of Italian citizens on the risk of infection by SARS-CoV-2 at the human–animal interface. The findings of the study showed that (a) the potential risk of SARS-CoV-2 transmission from humans to companion animals has been only partially perceived during the COVID-19 pandemic, (b) the knowledge of preventive measures to avoid SARS-CoV-2 transmission between humans and animals received limited attention by the Italian population, and (c) the communication campaign on COVID-19 and companion animals was, overall, considered inadequate in Italy. The main source of information for Italian citizens was represented by television broadcasts, while few Italians relied on veterinarians to obtain information on the risk of SARS-CoV-2 transmission between humans and companion animals. However, veterinarians were among the most trustworthy sources of information, suggesting that they and veterinary scientists in general could play a key role in the public communication of zoonoses and zoonotic pathogens.

**Abstract:**

This study analyzed data on the sources and the level of Italians’ awareness on the risk of infection by SARS-CoV-2 at the human–animal interface. Data were collected through a survey-type investigation on a representative sample of the Italian population. Forty-five percent of the interviewees were aware that companion animals could be infected by SARS-CoV-2. However, 29.8% were familiar with preventive measures to adopt to avoid viral transmission between infected humans and companion animals, and only 20.7% knew which companion animals could be at risk of infection. Higher awareness regarding the risk of SARS-CoV-2 transmission between animals and humans (51.7%) and the measures to prevent it (33.3%) was detected among companion animals’ owners. Notably, 40.4% of interviewees were not informed at all. Television broadcasts (26.4%) represented the main source of information, while only 3.5% of the interviewees relied on veterinarians, of which 31.9% considered this source of information as the most trustworthy. Overall, 72.4% of Italians recognized that the communication campaign on COVID-19 and companion animals was inadequate. This survey highlights the need for increasing the public awareness of the risk of companion animals being infected with SARS-CoV-2 and the involvement of professionals in the public communication on zoonoses.

## 1. Introduction

Starting from the first human cases of COVID-19 in the province of Hubei (China) in December 2019 [1,2], its causative agent, the severe acute respiratory syndrome coronavirus 2 (SARS-CoV-2), promptly spread worldwide. As the number of cases and the involvement of countries increased uncontrollably, the World Health Organization (WHO) declared COVID-19 a public health emergency of international concern in January 2020 and then a global pandemic in March 2020 (https://www.who.int/emergencies/diseases/novel-coronavirus-2019). At the time of writing this manuscript, the number of confirmed global cases and deaths exceeded 760 million and 6.8 million, respectively (https://covid19.who.int/ (accessed on 1 April 2023)). Italy was among the first countries to be affected by the COVID-19 pandemic, and in response to the health crisis, the state of emergency was declared on 31 January 2020 at national level (https://www.gazzettaufficiale.it/eli/id/2020/02/01/20A00737/sg). However, the first human cases of COVID-19 were identified in northern Italy at the end of February 2020, and today, the number of monthly confirmed cases is still above 97,000 (as of March 2023; https://www.iss.it (accessed on 1 April 2023)).

Soon after the first reports of human cases, domestic animals also, such as dogs and cats, were confirmed to be SARS-CoV-2 infected by molecular tests in Hong Kong and Belgium [3]. Over time, natural SARS-CoV-2 infections have been increasingly identified in a variety of animal species, showing or not clinical signs [4,5,6], such as nonhuman primates (i.e., gorillas and macaques), wild felids (i.e., lions, lynxes, tigers, pumas, snow leopards, and fishing cats), and other domestic and captive wild mammals (i.e., ferrets, hamsters, minks, white-tailed deer, otter, spotted hyenas, South American coati, hippopotamus, and binturong) [7,8]. Most animal cases have been reported in companion animals (in particular, cats and dogs) after natural exposure to SARS-CoV-2-positive owners [9,10], but dogs seem to be less commonly affected than cats [11]. Since infection of companion animals has been reported mainly after exposure to humans affected by COVID-19 [12,13,14], SARS-CoV-2 is now considered a reverse zoonotic agent [15,16], even though sporadic cases of animal-to-human transmission have also been described [17,18,19]. Indeed, the early human cases of SARS-CoV-2 have been linked back to strict contacts with wildlife in a market in Wuhan [20]; however, the animal reservoir of this virus is still uncertain, with bats and pangolins suspected to be the most probable candidates [21]. Neutralizing antibodies in dogs (2.3–3.3%) and cats (5.8–16.2%) living in close contact with SARS-CoV-2-positive humans in geographic areas severely affected by COVID-19 were described in Italian studies [10,22]. Furthermore, cases of mild or severe respiratory disease and asymptomatic infections have also been detected in Italian household cats and dogs [22,23,24]. However, evidence of infection by SARS-CoV-2 has not always been found in Italian pet animals [10].

Companion animals are commonly found in Italian households, accounting for 62.1 million pets in 2021, and this number seems to have increased continuously since 2017 (www.altroconsumo.it (accessed on 3 May 2022)). Indeed, it is reported that 62% of Italians own at least a dog and 56% at least a cat (www.altroconsumo.it (accessed on 3 May 2022)). In a recent survey, 69% and 71% of dog and cat owners, respectively, declared that companion animals improved their quality of life. During the COVID-19 pandemic, the contribution of companion animals to human wellbeing increased exponentially because of the strict containment measures (lockdown) repeatedly enforced in Italy, as well as in other countries, and helped citizens to cope with the stress and anxiety related to social isolation and uncertainty for the future [25,26]. At the same time, it is also necessary to stress the animals’ rights for their health and welfare to be guaranteed [27]. Indeed, the close and continuous cohabitation may have led to an increased risk of transmission of SARS-CoV-2 between owners and their companion animals. In this context, the knowledge and perception of the transmission risk between humans and animals could have played a significant role in increasing public awareness and the willingness to adopt preventive measures.

The COVID-19 pandemic not only represented a health emergency but also an emergency for public communication and information. The crisis and the related communication challenges involved all social actors and institutions, including the public ones, whose information efforts were not always perceived as effective by the citizens [28]. Furthermore, scientific experts faced an unprecedented situation, being continuously asked for opinions and public involvements. In 2022, the Science in Society Monitor (https://www.observa.it/en/category/monitor/ (accessed on 1 December 2022)) reported that over 80% of Italians had a high level of trust in scientific institutions and science in general. However, when considering the perceived public trustworthiness in scientific experts, a reduction to 43% was observed [29]. Media and citizens faced a critical knot of time that the health crisis made even more relevant—or, rather, that of the quality and credibility of information.

Considering this background, the aims of this survey were to (a) investigate the level of knowledge and awareness of Italian citizens on the risk of transmission of SARS-CoV-2 between humans and companion animals and (b) identify which sources the Italian population relied on to obtain the information on this risk and which sources were considered as the most trustworthy. Data collected in this study might help to understand the relationship between veterinary science and society and to develop a more effective communication on zoonoses and zoonotic pathogens by engaging veterinarians and veterinary scientists.

## 2. Materials and Methods

### 2.1. Survey and Target Population

A survey-type investigation was carried out, with the support of a specialized company, between 19 November and 8 December 2021 on 1008 Italian citizens aged ≥15 years, proportional and representative in terms of gender, age group, and geographical area of residence, by using the computer-assisted web interviewing (CAWI) technique and the computer-assisted telephone interview (CATI) technique in 70% and 30% of cases, respectively [30]. The population from which the sample was extracted was the Italian resident population aged ≥ 15 years with a landline phone or registered in the Opinioni.net (https://opinioni.net/) panel web (maximum margin of error at the 95% confidence level: 3.09%; response rate: 11.18%). The questionnaire included a first section of seven questions/items on the socio-demographic characteristics of the respondents and a second section including eight questions, one of which in a battery, for a total of eleven items on the relationship between humans and companion animals during the COVID-19 pandemic. In detail, the second section of the survey focused on (a) the participants’ knowledge and awareness of the risk of transmission of SARS-CoV-2 between humans and companion animals, (b) the main sources of information, and (c) which sources of information were most trustworthy. The questions/items of the survey are reported in Appendix A. On the data matrix obtained from the survey, data checking and weighting were carried out in order to obtain a proportional and representative sample of the Italian population in terms of level of education, as well as gender, age, and geographical area of residence. The sample obtained after weighting consisted of 985 cases (Table 1), among which 51.8% were females, 17.6% were between 15 and 29 years of age, 21.3% between 30 and 44 years of age, 26.5% between 45 and 59 years of age, and 34.6% ≥60 years of age. As in the Italian population, 52.5% of the sample had a low level of education (i.e., primary/secondary school), while 31.9% had a medium level of education (i.e., high school), and only 15.6% had a university degree.

Almost half (49.7%) of the Italian citizens were represented by companion animals’ owners, while only 8.2% declared they worked in close contact with animals. Gender, age, and level of education seemed to have an effect on companion animals’ ownership (Appendix A). Indeed, 53% of female respondents owned a companion animal vs. 46.1% of males. Furthermore, Italian citizens with a low level of education and older than 60 years of age were less likely to own a companion animal. Ethical review and approval were not required for the study on human participants in accordance with the local legislation and institutional requirements.

### 2.2. Statistical Analysis

Descriptive statistics and the Chi-square test were employed to analyze the level of knowledge and awareness of the Italian population of the risk of transmission of SARS-CoV-2 in the human–companion-animals relationship, the sources of information used by citizens, and their perception of trustworthiness and to assess the statistical significance of their bivariate relationship with the demographic characteristics of the participants (i.e., age, gender, level of education, geographical area, companion animal ownership, and working with animals). Furthermore, six logistic regression models were developed considering the demographic characteristics and the level of information on COVID-19 and companion animals as independent variables. In the models, the following items were included: (a) be aware that companion animals can also be infected with COVID-19; (b) be aware of the precautionary measures to be adopted when persons infected with COVID-19, or suspected of being infected with COVID-19, are in close contact with companion animals; (c) be aware of which companion animals are most at risk of COVID-19 infection; (d) believe that transmission of the virus can occur from companion animals to humans; (e) believe that humans can transmit the virus to companion animals; and (f) believe that several dogs and cats have become ill after close contact with ill people, as dependent variables. On the population sample that declared having searched for information on the risk of transmission (*n* = 586), six additional logistic regression models were computed by replacing the dichotomous variable, “Did you inform yourself about COVID-19 and companion animals?” with the variables “Where did you mainly find information about COVID-19 and companion animals?” and “Which source do you trust most for precautions to take?” Chi-square test and logistic regression models were carried out in SPSS Statistics version 28.0.1.1.

## 3. Results

### 3.1. Sources of Information and Public Communication on the Risk of Transmission of SARS-CoV-2 between Humans and Companion Animals among Italian Citizens

As retrieved by the questionnaire, 40.4% of the interviewed stated that they did not search for information on the risk for companion animals to be infected by SARS-CoV-2 (Table 2). Even though females were more likely to own companion animals, our survey showed that they were less prone (56.2%) than males (63.2%) to gain information on the risk of transmission of SARS-CoV-2 between humans and companion animals, whereas people below 30 years of age were shown to be more active in being informed (67.2%) (Appendix A). The main source of information was represented by television broadcasts (26.4%), followed by national newspapers (10.9%). Only 3.9% of Italians obtained information from social media. Interestingly, even though veterinarians were not among the primary sources of information (only 3.5% of Italians relied on them), they were considered the most trustworthy source of information (31.9%), followed by institutional websites (23.3%), such as that of the Italian Ministry of Health. The age and the level of education seemed to affect the source of information. In fact, people below 30 years of age acquired information from institutional websites (10.3%) and “friends and relatives” (12.1%), while among elderly people and those with a low level of education, television broadcasts represented the main source of information (39.3% and 46%, respectively). Newspapers (15.4%) and institutional websites (9.6%) were the main sources of information used by people possessing a high level of education. Not surprisingly, companion animals’ owners were more informed than people not owning any (65.8% and 53.3%, respectively). Moreover, only 6.3% of companion animals’ owners asked for information from his/her veterinarian. However, veterinarians were the most trusted according to citizens working with animals (47.4%), companion animals’ owners (43.5%), people with a low level of education (40.5%), elderly (41.9%), and females (36.9%).

Overall, more than 70% of Italian citizens declared that the communication regarding SARS-CoV-2 and companion animals was inadequate throughout the national awareness campaign on COVID-19. Elderly people (57.2%) were less in agreement with the statement that the public communication on SARS-CoV-2 and companion animals was inadequate, while companion animals’ owners (77.9%), people with a high level of education (83.2%), and those working with animals (75%) tended to agree more with this statement. Furthermore, among people relying on newspapers, institutional websites, “friends and relatives”, and those considering institutional websites and veterinarians as the most trustworthy sources of information, over 80% declared that the communication on SARS-CoV-2 and companion animals was inadequate.

### 3.2. Knowledge and Awareness of Italian Citizens on the Risk of Transmission of SARS-CoV-2 between Humans and Companion Animals

Almost 55% of the respondents declared not being aware that companion animals could be infected by SARS-CoV-2 and 79.3% declared not knowing which animal species could be at higher risk of infection after contact with infected humans (Table 2). The awareness on the possibility of companion animals to be infected with SARS-CoV-2 was higher in females (52%), companion animals’ owners (51.7%), and as the level of education increased (from 38.7% to 58.4%) (Appendix A). Eighty-three percent of people who did not inform themselves were not aware that companion animals could be infected by SARS-CoV-2. People aware of the potential risk of infection searched for information mainly on television broadcasts and newspapers (68.5% and 75.7%, respectively) and trusted the institutional websites and veterinarians (73% and 69.5%, respectively). Notably, among those who disagreed with the statement that communication on SARS-CoV-2 and companion animals was inadequate, 82% were not aware that the virus could infect companion animals.

Preventive measures to reduce the risk of transmission between infected owners and companion animals were known by only 29.8% of the Italian population. Companion animals’ owners (33.3%), young people (46.2%), and those working with animals (49.4%) seemed to be more aware of the preventive measures to adopt when COVID-19-positive humans are in strict contact with companion animals. The vast majority of uninformed people (92.2%) did not know which measures are necessary to reduce the risk of transmission from infected humans to companion animals. Young people (33.5%), people working with animals (37%), relying on veterinarians (41.2%) as a source of information, and trusting “friends and relatives” (66.7%) were more aware of which animal species were at higher risk of infection, in contrast to uninformed people (95.7%) and people considering public communication as adequate (93.1%).

In this survey, the knowledge of Italian citizens on the three items used in public discussion (i.e., SARS-CoV-2-positive companion animals can infect humans, SARS-CoV-2-positive humans can infect companion animals, and several dogs and cats are infected by SARS-CoV-2 after exposure to ill persons) was assessed (Table 2 and Appendix A). Overall, the elderly and people with the lowest level of education were those more likely to be unable to respond to the three items (23.4%, 29.2%, and 43.9%, respectively, for the elderly and 29.7%, 36.5%, and 49.3%, respectively, for people with the lowest level of education) (Table 2). In detail, 34.4% of respondents acknowledged that the virus could be transmitted from humans to animals, while about a quarter believed that infected companion animals could transmit the virus to humans (24.2%) and that several dogs and cats were infected after exposure to SARS-CoV-2-positive owners (25.6%) (Table 2). More than 72.0% of companion animals’ owners disagreed with the statement that SARS-CoV-2 transmission could occur from companion animals to humans. Lack of awareness of the risk of infection of companion animals (75.6%), the preventive measures to adopt to reduce such risk (71.5%), of which companion animals were more at risk of infection (72.2%), and lack of information (76.2%) were also common in those people in disagreement with the aforementioned statement. People relying on physicians and social media (42.1% and 57.9%, respectively), and those trusting physicians and “friends or relatives” (39.5% and 73.9%, respectively) as sources of information believed more than others that SARS-CoV-2 transmission could occur from companion animals to humans. More than 90% (93.1%) of people considering the public communication on this issue as inadequate did not believe that SARS-CoV-2 transmission could occur from companion animals to humans.

About 35.0% of companion animals’ owners and 43.8% of people working in strict contact with animals were aware that humans could infect companion animals. On the other hand, people unaware of the risk of infection of companion animals (68.4%), the preventive measures to adopt to reduce such risk (63.3%), and which companion animals were more at risk of infection (60.6%) did not know that companion animals could be infected by SARS-CoV-2 after exposure to COVID-19-positive humans. Similarly, people who did not search for information (69.9%), those trusting television broadcasts (65.0%), and those not believing that the communication was inadequate (93.1%) were also in disagreement with this item. Males (62.7%), people working in contact with animals (59.6%), unaware of the risk of infection of companion animals (68.6%), of the preventive measures to adopt to reduce such risk (64.9%), of which companion animals were more at risk of infection (63.4%), those who did not inform themselves (67.2%), and those not believing that the communication was inadequate (85.9%) were not aware that several dogs and cats tested positive for SARS-CoV-2 after exposure to infected humans. Those aware that several dogs and cats tested positive for SARS-CoV-2 after exposure to infected humans were young people (47.4%), possessed a high level of education (34.6%), and relied on institutional websites (53.1%) and social media of “friends and/or acquaintances” (48.6%) as sources of information and trusted “friends or relatives” (69.6%) on this issue.

### 3.3. Multiple Regression Results

Considering all the other independent variables, gender (female; *p* < 0.0001, O.R. = 2.400, 95% CI: 1.742–3.308), high level of education (degree; *p* < 0.0001, O.R. = 4.723, 95% CI 2.33–9.572), and being informed on the risk of transmission (*p* < 0.0001, O.R. = 11.697, 95% CI 8.129–16.831) were confirmed as variables influencing the awareness that companion animals could be infected by SARS-CoV-2 (Table 3). On the contrary, this model could not confirm the other significant correlations observed in the bivariate analysis, i.e., being companion animals’ owners and failing to be informed of the risk of transmission.

Awareness of the preventive measures to reduce the risk of transmission between infected owners and companion animals and of which companion animals were most at risk of infection was influenced by age (15–29 years of age *p* = 0.008, O.R. = 1.938, 95% CI 1.187–3.166 and *p* = 0.034, O.R. = 1.779, 95% CI 1.044–3.031, respectively), type of employment (*p* = 0.025, O.R. = 1.855, 95% CI 1.079–3.188 and *p* = 0.018, O.R. = 1.986, 95% CI 1.123–3.512, respectively), and opportunity to get information (*p* < 0.0001, O.R. = 9.073, 95% CI 5.935–13.871 and *p* < 0.0001, O.R. = 14.179, 95% CI 7.791–25.804, respectively) (Table 4).

Taking into account all the other dependent variables, young vs. old (15–29 years of age *p* < 0.0001, O.R. = 2.781, 95% CI 1.625–4.762), being informed (*p* < 0.0001, O.R. = 3.281, 95% CI 2.259–4.765), and considering the communication campaign on SARS-CoV-2 inadequate (*p* = 0.001, O.R. = 2.167, 95% CI 1.371–3.427) were positively correlated with the perception that positive companion animals could infect humans (Table 4). On the other hand, companion animals’ ownership was negatively correlated with this statement (*p* = 0.005, O.R. = 0.620, 95% CI 0.444–0.865). In addition to the correlations previously described, the item “SARS-CoV-2-positive humans can infect companion animals” was also influenced by the level of education (degree *p* = 0.005, O.R. = 3.406, 95% CI 1.447–8.017); in contrast, acknowledging that “several dogs and cats are infected by SARS-CoV-2 after exposure to ill persons” was not influenced by this variable (Table 4). On the other hand, working in strict contact with animals (*p* = 0.013, O.R. = 2.104, 95% CI 1.172–3.778) increased the awareness that companion animals had been infected by COVID-19-positive persons.

Based on the results of the multivariate analysis, being informed showed the highest odd ratio among all the other considered variables, demonstrating thus to be the most influential variable in the model. Therefore, the downstream analysis was focused on the population that acquired information on COVID-19 and companion animals by considering the main source of information and their perceived trustworthiness. In this model, gender and level of education were confirmed as significant variables influencing the awareness of the risk of infection for companion animals. Even though the trustworthiness in the source of information was not statistically significant, the source itself was significant. In detail, the model showed that people informing themselves from television and radio broadcasts (Table 5) were more likely to be aware that companion animals could be infected by SARS-CoV-2.

Regarding the item, “The transmission of the virus can occur from companion animals to humans”, age and the perception of communication on COVID-19 and companion animals as inadequate were confirmed as statistically significant, whereas companion animals’ ownership lost the net effect observed in the previous regression model (Table 6). While the source of information was not statistically significant, the trustworthiness showed an effect on the awareness of the Italian population. Remarkably, compared to people trusting the indications from institutions (e.g., Italian Ministry of Health, etc.), those trusting “friends and relatives” (*p* = 0.001, O.R. = 9.733, 95% CI 2.445–38.751) and “web forum or social media” (*p* < 0.0001, O.R. = 9.546, 95% CI 2.690–33.879) were more likely to believe that the transmission could occur from companion animals to humans (Table 6). Concerning the item, “Humans can transmit the virus to companion animals”, most of the statistically significant variables in the previous model (i.e., age, level of education, and inadequate communication on COVID-19) were confirmed, with the exception of companion animals’ ownership (Table 6). In respect to the source of information, the only significant difference was observed for the perceived trustworthiness of the physicians. Indeed, people trusting their own physicians (*p* = 0.043, O.R. = 2.103, 95% CI 1.024–4.318) were more prone to believe that companion animals could be infected by humans, compared to those trusting the suggestions provided by the institutions (Table 6).

Regarding the item, “Several dogs and cats became ill after close contact with ill people”, the considered variables (i.e., age, companion animals’ ownership, and working in strict contact with animals) were confirmed as statistically significant. Statistical significance was detected only for the trustworthiness of the source of information but not for the source of information (Table 6). In detail, people who trust “friends or relatives” (*p* = 0.024, O.R. = 3.998, 95% CI 1.197–13.356) and pharmacists (*p* = 0.004, O.R. = 5.957, 95% CI 1.763–20.131) more than the suggestions provided by the institutions were also more aware that several dogs and cats could be infected by SARS-CoV-2.

## 4. Discussion

In this survey, the source of information and the public perception regarding the communication on SARS-CoV-2 and companion animals was explored in the Italian population between November and December 2021. This survey was carried out almost two years after the beginning of the COVID-19 pandemic, when the scientific community was fully aware of the role played by wild and domestic animals, including companion animals, in the eco-epidemiology of the virus [4,14,21]. Considering that the transmission of microorganisms at the human–animal interface has been a well-recognized event since the early days of research on infectious diseases, soon after the onset of the COVID-19 pandemic, veterinary scientists promptly started investigating the risk of transmission of SARS-CoV-2 from humans to animals and vice versa. Therefore, their contribution was crucial in clarifying that humans played a major role in maintaining and transmitting the virus, while animals represented mainly accidental hosts of this human pathogen [31,32].

Regarding the possibility of SARS-CoV-2 transmission between humans and companion animals (i.e., “SARS-CoV-2-positive companion animals can infect humans”, “SARS-CoV-2-positive humans can infect companion animals”, “Several dogs and cats are infected by SARS-CoV-2 after exposure to ill person”), Italian citizens below 30 years of age and with a high level of education were shown to be more aware of these items compared to the elderly and people with a low level of education. The willingness and propensity of young Italian adults (<30 years old) to gain information on the risk of transmission of SARS-CoV-2 between humans and companion animals agrees with a previous study conducted in the US population [33]. Moreover, the lower inclination of older people to acquire, understand, and apply health information, mainly on zoonoses, compared to young adults was highlighted in the report, “The Health Literacy of America’s Adults” [34]. Our survey revealed that institutional websites (e.g., Italian Ministry of Health) were among the most trustworthy and accessed sources for gaining information on SARS-CoV-2 and companion animals by young and highly educated people, and this suggests that these official sources might be difficult to be reached by the elderly, thus reinforcing their unwillingness and/or inability to actively search and acquire information. Indeed, the main source of information for elderly and people with a low level of education was represented by television broadcasts, and on the other hand, they were the only respondents suggesting that the communication on SARS-CoV-2 and companion animals was adequate, and these results may suggest a lack of visibility of institutional representatives on television.

Of concern, the majority of the Italian population was not aware that companion animals, or any other animal, could be infected by SARS-CoV-2 at a time when several scientific papers reporting evidence of animal infections were already published [4,14,21] and the risk of infection for companion animals was common and accepted knowledge within the scientific community [6,10]. Our findings suggest that information on SARS-CoV-2 at the human–animal interface did not reach the public at large at the same speed and with the same effectiveness as it did within the scientific community, contrary to other COVID-19-related information. Indeed, the COVID-19 context was characterized by a high presence of—and public demand for—scientific experts and expertise in all aspects of citizens’ daily lives [35], and scientific communication was fundamental to reduce the transmission risk through information and guidelines shared on a variety of media but also provided a frame to understand and address this health crisis [28]. However, although guidelines on companion animals and SARS-CoV-2 published by the national health authorities were already publicly available, the vast majority of Italians were not aware of the preventive measures to adopt to reduce the risk of transmission of the virus from humans to companion animals. It is unclear whether this lack of awareness could be ascribed to the poor willingness of Italian citizens to search for information or to an inadequate dissemination of these guidelines.

Since the beginning of the COVID-19 pandemic, public communication played a key role both in framing the issue and in providing citizens with relevant information and guidelines on how to minimize the risk of transmission. Over 70% of Italian citizens stated that communication about SARS-CoV-2 and companion animals was inadequate and, as a result, over 90% of uninformed people were unaware of the preventive measures to mitigate the risk of transmission of SARS-CoV-2 between companion animals and their owners. As shown by logistic regression models, being informed had the highest odd ratio among all the other considered variables, demonstrating thus to be the most influential variable.

Overall, our survey showed that the majority of the interviewed Italian citizens were not aware and concerned about the risk for companion animals to be infected by SARS-CoV-2, about which species were at highest risk and the preventive measures to undertake to avoid virus transmission between humans and animals. Even though this finding might be perceived as worrying because of it suggesting scarce willingness to gain knowledge by the Italian population on the possibility of SARS-CoV-2 being transmitted between humans and companion animals, it is nevertheless in agreement with previous studies carried out in other European countries and worldwide [33,36]. However, in this discouraging scenario, we have revealed that companion animals’ owners were more informed than people not owning any pets; likewise, in a Californian survey, veterinarians declared that most of their clients expressed concern over the potential transmission of SARS-CoV-2 from their companion animals [37]. On the contrary, Powell et al. (2022) reported that about 80% of American pet owners were not concerned or only somewhat concerned about their pets contracting or transmitting the virus. Similarly, only 0.15% of British pet owners consulted their veterinarians to address their concerns over the possibility of their companion animals being infected by the COVID-19 virus [36]. Studies conducted across the 2010s in North America showed that pet owners believed veterinarians should be responsible for providing zoonotic disease information to the public and looked to veterinarians to gather information on this topic [38,39]. Similarly, more recent published surveys [33] and our survey showed that veterinarians were the most trusted source of information for the general public, suggesting that veterinary science, and therefore veterinarians and veterinary scientists, should play a key role in communications during health crises involving zoonotic pathogens. In the Italian public and communication context, the expertise of veterinarians working at different levels of public health was not fully recognized by the media, preventing them from reaching the public at large.

Thanks to changes in the media landscape, scientific experts have increasingly played a central role in public communication and in the broader relationships between science and society [40,41,42]. Unlike in other countries [43], in Italy the process of mediatization of science started since the onset of the COVID-19 pandemic, and many experts became familiar characters for Italians, including those holding official positions. Indeed, experts were assiduously approached by the media in search of other figures in addition to the official voices; however, only a few of them were veterinarians [44].

Together with the scarce public engagement, it should be also considered that during the COVID-19 pandemic, it was very difficult for Italian citizens to reach veterinarians (because of lockdown policy), except for emergencies. This limited access to veterinary care might have further prevented companion animals’ owners from relying on the advice of veterinarians. Indeed, our survey showed that the most common sources of information for Italian citizens on the COVID-19 pandemic were television broadcasts, which represented the most accessible and immediate sources of information for a population restricted to homes (lockdown) and with limited freedom of movement and social contacts. This finding is also in agreement with a previous survey [28] on public perception of COVID-19 vaccination of the Italian population, which showed television broadcasts to be the most common sources of information. The absence of veterinary scientists on media outlets might account for the poor effectiveness of the communication campaign on SARS-CoV-2 and companion animals, negatively influencing the public perception. Indeed, communication clarity and trust were among the most important issues that scientific experts had to address when engaged in public communication [45].

## 5. Conclusions

Overall, a common misperception of the topic “COVID-19 and companion animals” was observed in this survey. Indeed, interviewed Italian citizens demonstrated poor interest in gaining information on the potential risk of infection of companion animals and about which companion animals could be infected, as well as the preventive measures to adopt to reduce the risk. This misperception was not reported only in Italy but also in other countries during the first wave of the COVID-19 pandemic. A more adequate communication strategy should be implemented by directly involving professionals, who are perceived as the most scientifically reliable and trustworthy sources of information. This highlights the need to improve the scientific communication regarding the relationship between companion animals and humans and to develop more effective communication skills among veterinarian scientists and professionals and their institutions.

## Figures and Tables

**Table 1 vetsci-10-00426-t001:** Demographic characteristics of the interviewed Italian citizens (*n* = 985).

Demographic Characteristic	Frequency (n)	Percentage (%)
**Gender**		
Male	475	48.2
Female	510	51.8
**Age group**		
15–29 years	173	17.6
30–44 years	210	21.3
45–59 years	261	26.5
≥60	342	34.7
**Level of education**		
Primary school	138	14
Secondary school	380	38.5
High school	314	31.9
University degree	154	15.6
**Geographical area**		
Northwest	266	26.9
Northeast	225	22.9
Center	182	18.4
South	219	22.2
Islands	94	9.5

**Table 2 vetsci-10-00426-t002:** Relationship between humans and companion animals: monovariate distributions of responses from interviewed Italian citizens (*n* = 985).

Question	Frequency (n)		Percentage (%)		
**Do you own one or more companion animals?**					
No	496		50.3		
Yes	489		49.7		
**Do you work in strict contact with companion animals?**					
No	904		91.8		
Yes	81		8.2		
**Do you know that companion animals can be infected by SARS-CoV-2?**					
No	540		54.9		
Yes	445		45.1		
**Do you know the preventive measures to adopt when a COVID-19-positive person is in contact with a companion animal?**					
No	691		70.2		
Yes	294		29.8		
**Do you know which companion animals are more at risk of infection by SARS-CoV-2?**					
No	781		79.3		
Yes	204		20.7		
**Where did you find information on COVID-19 and companion animals?**					
I did not inform myself	398		40.4		
Television and/or radio broadcasts	260		26.4		
Newspapers (printed or online)	107		10.9		
Institutional websites (e.g., Italian Ministry of Health, etc.)	64		6.5		
Social media of friends and/or acquaintances	38		3.9		
Physicians	19		1.9		
Veterinarians	34		3.5		
Friends and/or relatives	53		5.3		
Other (unspecified)	12		1.2		
**Concerning the preventive measures to adopt with companion animals, which of these sources do you trust the most?**					
Indications from institutions (e.g., Italian Ministry of Health, etc.)	136		23.3		
Physicians	85		14.6		
Veterinarians	187		31.9		
Newspapers	34		5.8		
Television and/or radio broadcasts	81		13.8		
Friends and/or relatives	24		4		
Pharmacists	19		3.2		
Web forum and/or social media	20		3.4		
Total	586		100		
**How much do you agree with each of the following statements about COVID-19 and companion animals?**					
	**Not at all**	**Little**	**Enough**	**A lot**	**I do not know**
The transmission of the virus can occur from companion animals to humans	33.3	34	16.9	7.3	8.5
Humans can transmit the virus to companion animals	28.1	26.8	25.7	8.7	10.7
Several dogs and cats become ill after close contact with ill people	24.2	34.1	18.8	6.8	16.1
The communication on COVID-19 and companion animals has been inadequate	7.3	13.7	33.4	39	6.6

**Table 3 vetsci-10-00426-t003:** Multivariate logistic regression analysis showing the association between demographic characteristics of the interviewed Italian citizens and their awareness of SARS-CoV-2 infection in companion animals (*n* = 985).

Questions		*p*-Value	Odd Ratio (O.D.)	95% CI O.D.	
				Lower	Upper
**Are you aware that companion animals can also be infected by SARS-CoV-2?**					
Sex	Female = 1; Male = 0	<0.0001	2.400	1.742	3.308
Age	≥60	0.321			
	15–29	0.162	0.696	0.419	1.157
	30–44	0.100	0.666	0.410	1.081
	45–59	0.113	0.689	0.434	1.092
Level of education	Primary school	<0.0001			
	Secondary school	<0.0001	1.853	1.008	3.405
	High school	<0.0001	3.426	1.803	6.509
	University degree	<0.0001	4.723	2.330	9.572
	Companion animals ownership	0.207	1.230	0.891	1.699
	Working in strict contact with animals	0.906	0.966	0.549	1.701
	Information on COVID-19 and companion animals	<0.0001	11.697	8.129	16.831
	Communication on COVID-19 and companion animals was inadequate	0.052	1.478	0.997	2.192
	Constant	0.000	0.036		
**Are you aware of which preventive measures reduce the risk of transmission between SARS-CoV-2-infected owners and companion animals?**					
Sex	Female = 1; Male = 0	0.115	1.289	0.940	1.768
Age	≥60	<0.0001			
	15–29	<0.0001	1.938	1.187	3.166
	30–44	0.482	0.839	0.515	1.368
	45–59	0.431	0.830	0.522	1.320
Level of education	Primary school	0.416			
	Secondary school	0.141	1.600	0.856	2.990
	High school	0.228	1.494	0.777	2.873
	University degree	0.539	1.251	0.612	2.560
	Companion animals ownership	0.381	1.158	0.834	1.609
	Working in strict contact with animals	<0.0001	1.855	1.079	3.188
	Information on COVID-19 and companion animals	<0.0001	9.073	5.935	13.871
	Communication on COVID-19 and companion animals was inadequate	0.082	0.709	0.481	1.045
	Constant	0.000	0.059		
**Are you aware of which companion animals are most at risk of SARS-CoV-2 infection?**					
Sex	Female = 1; Male = 0	0.089	1.353	0.955	1.917
Age	≥60	<0.0001			
	15–29	<0.0001	1.779	1.044	3.031
	30–44	0.161	0.669	0.382	1.174
	45–59	0.325	0.768	0.453	1.301
Level of education	Primary school	0.170			
	Secondary school	0.039	0.508	0.267	0.965
	High school	0.210	0.652	0.334	1.273
	University degree	0.337	0.697	0.334	1.455
	Companion animals ownership	0.224	0.798	0.554	1.148
	Working in strict contact with animals	<0.0001	1.986	1.123	3.512
	Information on COVID-19 and companion animals	<0.0001	14.179	7.791	25.804
	Communication on COVID-19 and companion animals was inadequate	0.320	0.800	0.516	1.242
	Constant	0.000	0.061		

**Table 4 vetsci-10-00426-t004:** Multivariate logistic regression analysis showing the association between demographic characteristics of the interviewed Italian citizens and their awareness on the risk of SARS-CoV-2 transmission between humans and companion animals (*n* = 985).

Questions		*p*-Value	Odd Ratio (O.D.)	95% CI O.D.	
				Lower	Upper
**Are you aware that SARS-CoV-2-positive companion animals can infect humans?**					
Sex	Female = 1; Male = 0	0.141	1.274	0.923	1.758
Age	≥60	<0.0001			
	15–29	<0.0001	2.781	1.625	4.762
	30–44	<0.0001	1.972	1.159	3.356
	45–59	0.291	1.325	0.786	2.235
Level of education	Primary school	0.209			
	Secondary school	0.154	0.594	0.291	1.216
	High school	0.660	0.848	0.408	1.764
	University degree	0.470	0.749	0.342	1.641
	Companion animals ownership	<0.0001	0.620	0.444	0.865
	Working in strict contact with animals	0.485	0.808	0.445	1.469
	Information on COVID-19 and companion animals	<0.0001	3.281	2.259	4.765
	Communication on COVID-19 and companion animals was inadequate	<0.0001	2.167	1.371	3.427
	Constant	0.000	0.082		
**Are you aware that SARS-CoV-2-positive humans can infect companion animals?**					
Sex	Female = 1; Male = 0	0.158	1.252	0.917	1.710
Age	≥60	<0.0001			
	15–29	<0.0001	3.456	2.058	5.804
	30–44	<0.0001	2.034	1.244	3.327
	45–59	0.053	1.593	0.993	2.554
Level of education	Primary school	<0.0001			
	Secondary school	<0.0001	2.618	1.182	5.796
	High school	<0.0001	2.480	1.100	5.587
	University degree	<0.0001	3.406	1.447	8.017
	Companion animals ownership	<0.0001	0.621	0.449	0.858
	Working in strict contact with animals	0.392	1.287	0.722	2.295
	Information on COVID-19 and companion animals	<0.0001	5.066	3.571	7.185
	Communication on COVID-19 and companion animals was inadequate	<0.0001	2.658	1.744	4.050
	Constant	0.000	0.025		
**Are you aware that several dogs and cats were infected by SARS-CoV-2 after exposure to ill persons?**					
Sex	Female = 1; Male = 0	0.170	1.268	0.903	1.781
Age	≥60	<0.0001			
	15–29	<0.0001	4.217	2.314	7.685
	30–44	<0.0001	2.413	1.335	4.359
	45–59	0.052	1.776	0.995	3.171
Level of education	Primary school	0.059			
	Secondary school	0.448	0.730	0.324	1.646
	High school	0.848	1.084	0.477	2.462
	University degree	0.473	1.374	0.578	3.268
	Companion animals ownership	<0.0001	0.586	0.412	0.833
	Working in strict contact with animals	<0.0001	2.104	1.172	3.778
	Information on COVID-19 and companion animals	<0.0001	6.477	4.204	9.979
	Communication on COVID-19 and companion animals was inadequate	<0.0001	2.543	1.551	4.171
	Constant	0.000	0.028		

**Table 5 vetsci-10-00426-t005:** Multivariate logistic regression analysis showing the association among demographic characteristics of the interviewed Italian citizens, sources of information, their trustworthiness, and awareness of SARS-CoV-2 infection in companion animals (*n* = 586).

Questions		*p*-Value	Odd Ratio (O.D.)	95% CI O.D.	
				Lower	Upper
**Are you aware that companion animals can also be infected by SARS-CoV-2?**					
Sex	Female = 1; Male = 0	<0.0001	2.660	1.765	4.011
Age	≥60	0.920			
	15–29	0.882	1.051	0.545	2.028
	30–44	0.969	0.987	0.519	1.880
	45–59	0.623	0.860	0.471	1.569
Level of education	Primary school	<0.0001			
	Secondary school	<0.0001	4.057	1.897	8.673
	High school	<0.0001	7.014	3.102	15.857
	University degree	<0.0001	8.968	3.604	22.317
	Companion animals ownership	0.115	1.426	0.917	2.216
	Working in strict contact with animals	0.815	1.085	0.547	2.153
Where did you find information on COVID-19 and companion animals?	Television and/or radio broadcasts	<0.0001			
	Newspapers (printed or online)	0.493	0.811	0.445	1.476
	Institutional websites (e.g., Italian Ministry of Health, etc.)	<0.0001	0.358	0.185	0.694
	Social media of friends and/or acquaintances	<0.0001	0.335	0.144	0.782
	Physicians	<0.0001	0.267	0.091	0.780
	Veterinarians	<0.0001	0.229	0.096	0.545
	Friends and/or relatives	<0.0001	0.230	0.110	0.480
	Other (unspecified)	0.105	4.846	0.719	32.682
Concerning the preventive measures to adopt with companion animals, which of these sources do you trust the most?	Indications from institutions (e.g., Italian Ministry of Health, etc.)	0.271			
	Physicians	0.138	0.597	0.302	1.180
	Veterinarians	0.866	0.949	0.517	1.741
	Newspapers	0.666	1.241	0.466	3.305
	Television and/or radio broadcasts	0.041	0.475	0.233	0.968
	Friends and/or relatives	0.777	1.178	0.379	3.660
	Pharmacists	0.325	0.580	0.196	1.714
	Web forum or social media	0.517	0.687	0.221	2.137
	The communication on COVID-19 and companion animals was inadequate	0.251	1.349	0.809	2.249
	Constant	0.013	0.339		
**Are you aware of which preventive measures reduce the risk of transmission between SARS-CoV-2-infected owners and companion animals?**					
Sex	Female = 1; Male = 0	<0.0001	1.776	1.224	2.577
Age	≥60	<0.0001			
	15–29	0.075	1.711	0.947	3.091
	30–44	0.079	0.595	0.333	1.062
	45–59	0.167	0.683	0.398	1.174
Level of education	Primary school	0.422			
	Secondary school	0.137	1.716	0.842	3.497
	High school	0.246	1.558	0.737	3.292
	University degree	0.530	1.301	0.573	2.956
	Companion animals ownership	0.070	1.448	0.970	2.162
	Working in strict contact with animals	<0.0001	2.280	1.179	4.408
Where did you find information on COVID-19 and companion animals?	Television and/or radio broadcasts	<0.0001			
	Newspapers (printed or online)	0.123	0.663	0.394	1.117
	Institutional websites (e.g., Italian Ministry of Health, etc.)	0.004	2.484	1.329	4.642
	Social media of friends and/or acquaintances	0.914	1.046	0.464	2.355
	Physicians	0.598	1.333	0.458	3.881
	Veterinarians	0.136	1.858	0.823	4.195
	Friends and/or relatives	0.449	0.764	0.380	1.535
	Other (unspecified)	0.407	0.556	0.139	2.226
Concerning the preventive measures to adopt with companion animals, which of these sources do you trust the most?	Indications from institutions (e.g., Italian Ministry of Health, etc.)	0.223			
	Physicians	0.366	0.742	0.389	1.417
	Veterinarians	0.307	0.760	0.449	1.287
	Newspapers	0.709	1.175	0.504	2.740
	Television and/or radio broadcasts	0.341	0.728	0.379	1.399
	Friends and/or relatives	<0.0001	0.279	0.094	0.833
	Pharmacists	0.826	0.886	0.301	2.609
	Web forum or social media	0.252	1.943	0.624	6.052
	The communication on COVID-19 and companion animals was inadequate	0.553	0.863	0.530	1.404
	Constant				
**Are you aware of which companion animals are most at risk of SARS-CoV-2 infection?**					
Sex	Female = 1; Male = 0	0.464	1.154	0.787	1.694
Age	≥60	<0.0001			
	15–29	0.048	1.838	1.004	3.363
	30–44	0.309	0.721	0.384	1.353
	45–59	0.519	0.824	0.458	1.483
Level of education	Primary school	0.478			
	Secondary school	0.185	0.613	0.298	1.263
	High school	0.425	0.734	0.344	1.567
	University degree	0.686	0.843	0.369	1.927
	Companion animals ownership	0.295	0.797	0.522	1.218
	Working in strict contact with animals	<0.0001	2.096	1.111	3.955
Where did you find information on COVID-19 and companion animals?	Television and/or radio broadcasts	0.634			
	Newspapers (printed or online)	0.589	0.862	0.502	1.479
	Institutional websites (e.g., Italian Ministry of Health, etc.)	0.189	0.633	0.320	1.253
	Social media of friends and/or acquaintances	0.268	0.594	0.236	1.494
	Physicians	0.964	0.975	0.324	2.938
	Veterinarians	0.394	1.420	0.634	3.181
	Friends and/or relatives	0.186	0.616	0.301	1.263
	Other (unspecified)	0.735	0.781	0.186	3.281
Concerning the preventive measures to adopt with companion animals, which of these sources do you trust the most?	Indications from institutions (e.g., Italian Ministry of Health, etc.)	<0.0001			
	Physicians	0.236	0.666	0.340	1.305
	Veterinarians	0.989	1.004	0.580	1.737
	Newspapers	0.051	2.300	0.997	5.307
	Television and/or radio broadcasts	<0.0001	0.433	0.198	0.945
	Friends and/or relatives	<0.0001	3.428	1.226	9.580
	Pharmacists	0.574	1.362	0.464	3.999
	Web forum or social media	0.415	1.571	0.530	4.653
	The communication on COVID-19 and companion animals was inadequate	0.581	0.866	0.520	1.442
	Constant	0.659	0.835		

**Table 6 vetsci-10-00426-t006:** Multivariate logistic regression analysis showing the association among demographic characteristics of the interviewed Italian citizens, sources of information, their trustworthiness, and awareness of the risk of SARS-CoV-2 transmission between humans and companion animals (*n* = 586).

Questions		*p*-Value	Odd Ratio (O.D.)	95% CI O.D.	
				Lower	Upper
**Are you aware that SARS-CoV-2-positive companion animals can infect humans?**					
Sex	Female = 1; Male = 0	0.183	1.324	0.876	2.002
Age	≥60	<0.0001			
	15–29	<0.0001	2.184	1.103	4.326
	30–44	0.096	1.765	0.904	3.448
	45–59	0.977	0.990	0.508	1.931
Level of education	Primary school	0.200			
	Secondary school	0.083	0.463	0.194	1.104
	High school	0.427	0.693	0.280	1.715
	University degree	0.268	0.576	0.217	1.529
	Companion animals ownership	0.112	0.697	0.446	1.087
	Working in strict contact with animals	0.109	0.535	0.249	1.150
Where did you find information on COVID-19 and companion animals?	Television and/or radio broadcasts	0.146			
	Newspapers (printed or online)	0.558	0.835	0.456	1.528
	Institutional websites (e.g., Italian Ministry of Health, etc.)	0.059	1.877	0.977	3.605
	Social media of friends and/or acquaintances	0.214	1.756	0.722	4.270
	Physicians	0.360	1.739	0.532	5.688
	Veterinarians	0.423	1.458	0.579	3.669
	Friends and/or relatives	0.295	1.465	0.716	2.998
	Other (unspecified)	0.097	0.043	0.001	1.765
Concerning the preventive measures to adopt with companion animals, which of these sources do you trust the most?	Indications from institutions (e.g., Italian Ministry of Health, etc.)	<0.0001			
	Physicians	0.248	1.507	0.752	3.021
	Veterinarians	0.250	0.701	0.383	1.283
	Newspapers	<0.0001	2.739	1.092	6.869
	Television and/or radio broadcasts	0.621	1.198	0.584	2.457
	Friends and/or relatives	<0.0001	9.733	2.445	38.751
	Pharmacists	<0.0001	4.071	1.285	12.892
	Web forum or social media	<0.0001	9.546	2.690	33.879
	The communication on COVID-19 and companion animals was inadequate	<0.0001	2.805	1.515	5.196
	Constant	0.002	0.202		
**Are you aware that SARS-CoV-2-positive humans can infect companion animals?**					
Sex	Female = 1; Male = 0	0.244	1.263	0.853	1.870
Age	≥60	<0.0001			
	15–29	<0.0001	5.603	2.868	10.947
	30–44	<0.0001	2.834	1.533	5.241
	45–59	<0.0001	2.175	1.216	3.889
Level of education	Primary school	<0.0001			
	Secondary school	<0.0001	4.121	1.574	10.787
	High school	<0.0001	2.900	1.073	7.835
	University degree	<0.0001	4.647	1.621	13.320
	Companion animals ownership	0.221	0.764	0.497	1.175
	Working in strict contact with animals	0.112	1.860	0.866	3.996
Where did you find information on COVID-19 and companion animals?	Television and/or radio broadcasts	0.492			
	Newspapers (printed or online)	0.604	1.156	0.668	1.999
	Institutional websites (e.g., Italian Ministry of Health, etc.)	0.711	1.129	0.595	2.142
	Social media of friends and/or acquaintances	0.716	1.166	0.510	2.665
	Physicians	0.452	0.653	0.215	1.984
	Veterinarians	0.828	0.911	0.392	2.117
	Friends and/or relatives	0.228	1.613	0.741	3.511
	Other (unspecified)	0.065	0.195	0.034	1.106
Concerning the preventive measures to adopt with companion animals, which of these sources do you trust the most?	Indications from institutions (e.g., Italian Ministry of Health, etc.)	0.188			
	Physicians	<0.0001	2.103	1.024	4.318
	Veterinarians	0.398	1.272	0.728	2.221
	Newspapers	0.122	2.075	0.822	5.237
	Television and/or radio broadcasts	0.713	0.879	0.441	1.750
	Friends and/or relatives	0.801	1.158	0.370	3.620
	Pharmacists	0.879	1.095	0.341	3.518
	Web forum or social media	0.221	0.501	0.166	1.516
	The communication on COVID-19 and companion animals was inadequate	<0.0001	2.043	1.192	3.501
	Constant	0.000	0.062		
**Are you aware that several dogs and cats were infected by SARS-CoV-2 after exposure to ill persons?**					
Sex	Female = 1; Male = 0	0.141	1.355	0.904	2.033
Age	≥60	<0.0001			
	15–29	<0.0001	5.575	2.756	11.280
	30–44	<0.0001	2.943	1.483	5.844
	45–59	<0.0001	2.475	1.269	4.827
Level of education	Primary school	0.345			
	Secondary school	0.700	1.198	0.478	3.006
	High school	0.544	1.340	0.520	3.453
	University degree	0.185	1.969	0.724	5.355
	Companion animals ownership	<0.0001	0.529	0.340	0.825
	Working in strict contact with animals	<0.0001	2.529	1.246	5.131
Where did you find information on COVID-19 and companion animals?	Television and/or radio broadcasts	0.582			
	Newspapers (printed or online)	0.689	0.888	0.496	1.589
	Institutional websites (e.g., Italian Ministry of Health, etc.)	0.506	1.247	0.651	2.390
	Social media of friends and/or acquaintances	0.629	0.806	0.336	1.934
	Physicians	0.824	0.880	0.286	2.710
	Veterinarians	0.555	0.770	0.324	1.832
	Friends and/or relatives	0.110	0.529	0.242	1.155
	Other (unspecified)	0.184	0.305	0.053	1.760
Concerning the preventive measures to adopt with companion animals, which of these sources do you trust the most?	Indications from institutions (e.g., Italian Ministry of Health, etc.)	<0.0001			
	Physicians	0.093	1.827	0.904	3.692
	Veterinarians	0.155	1.517	0.854	2.693
	Newspapers	0.346	0.621	0.231	1.671
	Television and/or radio broadcasts	0.607	1.211	0.583	2.517
	Friends and/or relatives	<0.0001	3.998	1.197	13.356
	Pharmacists	<0.0001	5.957	1.763	20.131
	Web forum or social media	0.644	1.301	0.427	3.969
	The communication on COVID-19 and companion animals was inadequate	<0.0001	2.458	1.343	4.498
	Constant	0.000	0.088		

## Data Availability

The data that support the findings of this study are available from the corresponding author upon reasonable request.

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
