# Peer review of "SARS-CoV-2 and Companion Animals: Sources of Information and Communication Campaign during the COVID-19 Pandemic in Italy"

_vetsci, 2023, doi:10.3390/vetsci10070426_

Round 1
Reviewer 1 Report
I have no suggestions to improve the article.
The article is well written. A few minor corrections are necessary to make it adequate for publication.
Author Response
I have no suggestions to improve the article.
The article is well written. A few minor corrections are necessary to make it adequate for publication.
- The Authors wish to thank the reviewer and made some corrections in the manuscript.
Reviewer 2 Report
The authors' manuscript is very interesting and well presented. The only clarification and addition I would ask for is to specify how the sample size was calculated in order to define it as representative.
I would also like to ask you to specify how the respondents were selected and reached in order to make the survey replicable. Were they selected randomly? Was the sample size adjusted to account for deviations from simple random sampling?
Line 120-121 please provide references
Line 364-370 I suggest that the sentences should be reformulated in more than one sentence.
Thank you for sharing your work and for your valuable contribution to the scientific community.
Best wishes to all
Minor editing of English language required
Author Response
The authors' manuscript is very interesting and well presented. The only clarification and addition I would ask for is to specify how the sample size was calculated in order to define it as representative.
I would also like to ask you to specify how the respondents were selected and reached in order to make the survey replicable. Were they selected randomly? Was the sample size adjusted to account for deviations from simple random sampling?
- The research team decided to administer the questionnaire to a sample of 1,000 cases of the Italian population aged 15 years and over, proportional and representative in terms of gender, age group and geographical area of residence. With the support of a specialised company, the questionnaire was administered between 19 November and 8 December 2021 using the CAWI (Computer Assisted Web Interviewing) technique in 70% of cases and the CATI (Computer Assisted Telephone Interview) technique in 30% of cases (Marsden, P.V.; Wright, J.D. Handbook of survey research, 2nd Edition, Thousand Oaks (CA), Sage, 2010). The population from which the sample was extracted is the Italian resident population aged over 15 years with a landline phone or registered in the Opinioni.net panel web. Maximum margin of error (at the 95% confidence level): 3.09%. Response rate: 11.18%. On the data matrix obtained from the survey, the research team carried out data checking and weighting so that the sample was also representative by level of education, as well as by gender, age group and geographical area of residence. The distribution of the four variables used for the construction of the sample was obtained from I.Stat. Paragraph 2.1 of the manuscript had been modified accordingly.
Line 120-121 please provide references
- The following refernce was added to the manuscript: “Marsden, P.V.; Wright, J.D. Handbook of survey research, 2nd Edition, Thousand Oaks (CA), Sage, 2010.”
Line 364-370 I suggest that the sentences should be reformulated in more than one sentence.
- Revised as suggested.
Thank you for sharing your work and for your valuable contribution to the scientific community.
Best wishes to all.
- Thank you for your revision and your valuable comments.
Minor editing of English language required
- Revised as suggested.
Reviewer 3 Report
Dear authors,
Please consider the following changes,
and improve the literature review with recent references.
the introduction is confusing which makes the reader easily distracted from the content.
What is the study's primary goal, and what does this add to science and practicality? Add this to the introduction.
The results section should also be improved, together with the discussion and practical implications.
Author Response
Dear authors,
Please consider the following changes, and improve the literature review with recent references.
- The Authors wish to thank the reviewer who provided valuable comments for improving the manuscript. Accordingly, the authors have updated the references by adding newest ones.
the introduction is confusing which makes the reader easily distracted from the content.
- The Authors have revised the introduction in order to make it clearer and easier to read.
What is the study's primary goal, and what does this add to science and practicality? Add this to the introduction.
- From line 113 to line 120 (original version of the manuscript), the goals of the study and their added value to science and practicality are described. If the reviewer believes that the section should be expanded, the Authors are willing to comply with the request.
The results section should also be improved, together with the discussion and practical implications.
The Authors have revised the manuscript accordingly.
Reviewer 4 Report
Review of Laconi et al.
In this study, the authors performed survey-type investigation and analyzed Italians’ awareness on the risk of infection by SARS-CoV-2 at the human-animal interface. They found most of the interviewees were aware that companion animals could be infected with SARS-CoV-2, while still some portion of people did not. They also pointed out most interviewees believed the communication campaign on COVID-19 and companion animals was inadequate. This paper highlighted the need for increasing the public awareness on the risk of companion animals to be infected with SARS-CoV-2 with professionals get more involved. This work in general is interesting, while there are still some minor issues with data analysis need to be addressed.
1. This survey was done at the end of 2021, which was long time ago, why the authors trying to publish this data more than one year later? Will this affect the conclusions in this work?
2. Do the authors know what is the percentage of COVID-19 patients are infected by companion animals?
3. Beyond Italy, can the authors compare their data to other countries as well (beside America)? By doing so, are these data consistent with each other? What would be the major factor(s) if they are not?
4. Can the authors comment on what would be the best way(s) to increase the public awareness of this risk?
5. “Sars-CoV-2” should change to “SARS-CoV-2”.
See comments
Author Response
In this study, the authors performed survey-type investigation and analyzed Italians’ awareness on the risk of infection by SARS-CoV-2 at the human-animal interface. They found most of the interviewees were aware that companion animals could be infected with SARS-CoV-2, while still some portion of people did not. They also pointed out most interviewees believed the communication campaign on COVID-19 and companion animals was inadequate. This paper highlighted the need for increasing the public awareness on the risk of companion animals to be infected with SARS-CoV-2 with professionals get more involved. This work in general is interesting, while there are still some minor issues with data analysis need to be addressed.
- The Authors wish to thank the reviewer who appreciated the study and provided valuable comments for improving the manuscript.
- This survey was done at the end of 2021, which was long time ago, why the authors trying to publish this data more than one year later? Will this affect the conclusions in this work?
- The Authors agree with the reviewer’s comment. Unfortunately, due to the huge amount of data collected while the COVID-19 emergency was still dramatically ongoing in Italy, it was not possible to analyze those data promptly after their collection. Despite these challenges, the Authors kept going ahead with the data analysis and publication, since they were convinced that sharing these findings with the scientific community was however valuable, also because of the important role of the veterinarians and veterinary science research discovered by the survey. To conclude, the Authors would like to point out that unfortunately Sars-CoV-2 and companion animals have remained a neglected subject in the Italian (and all over the world) communication campaing and debate on COVID-19 since that time, thus the findings of the study are still of great value for the scientific community and the population at large.
- Do the authors know what is the percentage of COVID-19 patients are infected by companion animals?
- From line 73 to line 74 (original version of the manuscript), the Authors stated that companion animals to humans transmission is sporadic and only two reports were available at the time of writing; a further search the Authors have performed after the reviewers’ comment, confirmed that no new prevalence data have been published after the submission of the manuscript. However, the Authors also believe that this information is not essential to understand and appreciate the manuscript.
- Beyond Italy, can the authors compare their data to other countries as well (beside America)? By doing so, are these data consistent with each other? What would be the major factor(s) if they are not?
- The Authors would like to compare the results of this survey; however, only few studies have been published on this topic. Unfortunately, the further search the Authors have performed after the reviewers’ comment has not yield any new evidence regarding this issue.
- Can the authors comment on what would be the best way(s) to increase the public awareness of this risk?
- As described in the manuscript, it is the Authors’ opinion that the best way to increase public awareness would be to include experts, such as veterinarians and veterinary scientists, in public discussions on COVID-19.
- “Sars-CoV-2” should change to “SARS-CoV-2”.
- Revised as suggested.